# Cherenkov Radiation Detection on a LS Counter for ^226^Ra Determination in Water and Its Comparison with Other Common Methods

**DOI:** 10.3390/ma14216719

**Published:** 2021-11-08

**Authors:** Ivana Stojković, Nataša Todorović, Jovana Nikolov, Branka Radulović, Michele Guida

**Affiliations:** 1Faculty of Technical Sciences, University of Novi Sad, Trg Dositeja Obradovića 6, 21000 Novi Sad, Serbia; ivana_st@uns.ac.rs; 2Department of Physics, Faculty of Sciences, University of Novi Sad, Trg Dositeja Obradovića 3, 21000 Novi Sad, Serbia; jovana.nikolov@df.uns.ac.rs (J.N.); branka.radulovic@df.uns.ac.rs (B.R.); 3Department of Computer Engineering, Electrical Engineering and Applied Mathematics (DIEM), University of Salerno, 84084 Fisciano, Italy; miguida@unisa.it; 4Institute of Construction Technology, Faculty of Civil Engineering, Riga Technical University, 1658 Riga, Latvia

**Keywords:** ^226^Ra in water, Cherenkov radiation, liquid scintillation counting, gamma-spectroscopy, Quantulus 1220^TM^

## Abstract

Reliable determination of ^226^Ra content in drinking water, surface water and groundwater is required for radiological health-risk assessment of populations and radiation-dose calculations after ingestion and inhalation. This study aimed to determine ^226^Ra presence in the untreated water samples on a liquid scintillation counter via Cherenkov radiation detection. Cherenkov counting is a faster, simpler, less expensive technique than other commonly used methods for ^226^Ra determination. Step-by-step optimization of this technique on the Quantulus detector is presented in this paper. Improvement of detection limit/efficiency in the presence of sodium salicylate was investigated in this study. The main parameters of the method obtained were detection efficiency 15.87 (24)% and detection limit 0.415 Bq/L achieved for 1000 min of counting in 20 mL of sample volume. When 1 g of sodium salicylate was added, efficiency increased to 38.1 (5)%, with a reduction in the detection limit to 0.248 Bq/L for 500 min of counting. A satisfactory precision level of Cherenkov counting was obtained, the results deviating between 5% and 20% from reference values. The precision and accuracy of the Cherenkov counting technique were compared to liquid scintillation counting (EPA Method 913.0 for radon determination) and gamma spectrometry (the direct method for the untreated water samples on HPGe spectrometer). An overview of the advantages/disadvantages of each technique is elaborated in this paper.

## 1. Introduction

Drinking water may contain radioactive isotopes that pose potential risks to human health. Isotopes from the primordial uranium-238 series are the predominant contribututors to irradiation risks due to the ingestion of drinking water [1]. It is important to detect radium (^226^Ra) presence in natural water samples, because it is one of the most hazardous naturally occurring radionuclides concerning internal radiation exposure. Determination of ^226^Ra in natural water is needed to assess the dose due to ingestion and the properties of ^226^Ra deposition in the bones and the urinary tract. It has been demonstrated [2] that mortality rate due to bone cancer significantly increased in the areas where tap water contained ^226^Ra in concentrations greater than 110 mBq/L. In the study [3], increased rates of bladder carcinoma in men, breast cancer in women and lung cancer in both sexes were recorded with increasing concentrations of ^226^Ra in drinking water. Research into the incidence of leukemia [4] showed correlation of the disease with ^226^Ra activity concentration higher than 185 mBq/L in the ground water. Therefore, in monitoring studies, it is desirable to develop a precise and accurate technique for the determination of the activity concentration of this radionuclide [5]. The permitted activity concentration of ^226^Ra in drinking water according to Serbian legislation is 0.49 Bq/L [6]. The international guidance level for naturally occurring ^226^Ra content in drinking water is set to 1 Bq/L, according to the World Health Organization [7].

^226^Ra can be detected directly via its α-particle or γ-ray emission. Another way is indirect measurement of the activity of its progenies where radioactive equilibrium is required: α-particles (emitted from ^222^Rn, ^218^Po, ^214^Po), β-particles (emitted from ^214^Pb, ^214^Bi) and γ-ray emitters (again ^214^Pb, ^214^Bi) allow indirect determination of ^226^Ra [8]. The EPA (Environmental Protection Agency) has approved 17 methods for ^226^Ra analysis in drinking water [9]. Seven of the approved methods use a radiochemical/precipitation methodology to measure the total soluble alpha-emitting radioisotopes of radium, namely, ^223^Ra, ^224^Ra and ^226^Ra; ten of the methods use a radon-emanation methodology that is specific to ^226^Ra. The radiochemical methods do not always give an accurate measurement of ^226^Ra content when other radium emitters are present, but can be used for the screening of the samples [9]. 

There have been few recent attempts in the literature to evaluate and compare numerous analytical methodologies for radium determination [10,11,12]. One study [10] evaluated gamma spectrometry, liquid scintillation counting (LSC) and alpha spectrometry for radium measurements in environmental samples, concluding that α-spectrometry coupled with chemical separation offered maximal sensitivity with a detection limit of ~0.1 mBq/L (approximately two orders of magnitude lower than low-background HPGe γ-spectrometry and LSC techniques). For monitoring purposes in water samples, α-particle spectrometry was determined as the most suitable technique for ^226^Ra measurements [12]. The latest study [11] determined that LSC spectrometry coupled with extractive techniques and alpha-beta discrimination offers the most accurate, rapid and relatively simple determination of ^226^Ra activity.

This paper presents an exploration of the Cherenkov counting technique on an LS counter, a method that has not been widely used for radium determination so far. The advantages of Cherenkov counting over common LSC methods are: lower background count-rates and consequently lower detection limits, non-usage of expensive, environmentally unfriendly LS cocktails, and, consequently, simpler sample preparation with environmentally friendly disposal [13,14]. It has been documented that Cherenkov counting can be used for detection of hard beta-emitting radionuclides via LSC, but its counting efficiency is sensitive to color quench, and depends on the emitted β-energy, the sample volume and its concentration, the type of counting vial, rthe efractive index and the type of photocathode [15]. The motivation for the experiments presented in this paper was the lack of exhaustive data in the literature concerning the optimization of LS counters and the reliability of Cherenkov radiation detection for the purpose of ^226^Ra activity measurements. The uniqueness of this research lies in the fact that scientific literature did not introduce exact data on detection limits and techniques for its reduction in the case of ^226^Ra measurement via Cherenkov counting. Therefore, this paper offers a novel, extensive analysis of Cherenkov counting via LS counter: a step-by-step optimization of the Quantulus 1220^TM^ detector with an evaluation of the main parameters, such as selected spectral window, background count-rate, Minimal Detectable Activity *MDA* and detection efficiency. Results obtained in this research will complement to a large degree the existing experimental data concerning the relevance of the Cherenkov counting technique. The aim of the paper was to investigate the detection of ^226^Ra in water via Cherenkov radiation detection for monitoring of untreated water samples, and for that purpose, calibration samples and intercomparison samples were prepared with distilled water spiked with ^226^Ra solution. The results displayed in this research will supplement scientific literature with explicit and conclusive data on the possibilities, limitations and upgrades of the Cherenkov counting technique with regard to ^226^Ra determination in water using the Quantulus LS counter. The possible problem of interference by other radionuclides capable of generating Cherenkov radiation should be addressed in future work, and could involve pretreatment of water samples so that the presence of other radionuclides is eliminated.

Furthermore, we report that significant improvement in detection efficiency, and consequently lower detection limits, were achieved with the addition of sodium salicylate to the counting vial. The addition of sodium salicylate as a wavelength-shifter had been confirmed to increase the efficiency of Cherenkov counting in the case of ^228^Ra/^228^Ac [13] and ^210^Pb/^210^Bi [14] detection. The novelty of our research represents an investigation of the effects of sodium salicylate on ^226^Ra detection as well.

The second aim of the paper was to compare the precision and accuracy of the Cherenkov counting technique to two other commonly used methods for ^226^Ra determination in water: LSC and gamma spectrometry. The validity and performance of the analytical method can be appropriately examined via samples that contain known concentrations of ^226^Ra standard solution [11]. Therefore, intercomparison samples have been prepared with distilled water spiked with different concentrations of ^226^Ra isotope solution. 

For ^226^Ra determination by LSC, EPA Method 913.0 for radon determination in drinking water [16] was tested. Many procedures for determining ^226^Ra activity concentrations in water involve the determination of ^222^Rn, its daughter product (either alone or together with its other daughter nuclides) by LSC techniques; therefore, any measurement of ^226^Ra will also be relevant to ^222^Rn. Lastly, ^226^Ra in water samples was determined by gamma spectrometry using the direct method on an HPGe spectrometer (the untreated water samples). The results of the presented experiments provide the basis for discussion on the performance, precision and accuracy of each method. Therefore, the second objective of the paper was to offer a survey of the comparative advantages and disadvantages of two other frequently utilized techniques for ^226^Ra determination in water.

## 2. Materials and Methods

### 2.1. Cherenkov Counting Method and Materials Used

To optimize the measurement method and to establish its main parameters, the counting of a set of calibration samples on the detector was carried out. The obtained measurements were used to determine the optimal spectral window (ROI), the detection efficiency and to conduct a Minimal Detectable Activity (*MDA*) evaluation. The detection efficiency *ε* was obtained from the following expression:(1)$$\varepsilon =\frac{r_{C}-r_{0}}{C_{C}}$$
where *C_C_* [Bq] represents the reference activity of the ^226^Ra calibration sample (reference standard), and *r_C_* [cps] and *r*_0_ [cps] are the count-rates of the calibration sample and the background sample, respectively. Once the detection efficiency was established, the unknown sample activity concentration *C* [Bq/L] could be obtained as:(2)$$C=\frac{r_{S}-r_{0}}{V\;\varepsilon }$$
where *V* [L] and *r_S_* [cps] represent the analyzed volume and the count-rate of the water sample, respectively. The Currie relation can be used for the Minimal Detectable Activity *MDA* [Bq/L] parameter assessment [17]:(3)$$MDA=\frac{2.71+4.65\sqrt{r_{0}\;t_{0}}}{V\;\varepsilon \;t_{0}}$$
where *t*_0_ [s] represents the background counting time.

Cherenkov background rate varies with the vial type and the total sample volume. All experiments were performed using low-diffusion polyethylene vials (Super PE vial Cat.No. 6008117, PerkinElmer, Turku, Finland). Plastic vials transmit more light from Cherenkov radiation than glass vials [15]. Alternatively, glass vials could be used, but their ^40^K content would yield 2–3 times more background [18]. The sample volume was fixed at maximal vial capacity, 20 mL, since the Currie relation suggests that the increase in the analyzed volume proportionally reduces detection limits, Equation (3).

Cherenkov radiation was detected on an Ultra Low Level Liquid Scintillation Spectrometer Wallac 1220^TM^ Quantulus manufactured by PerkinElmer Life Sciences (Turku, Finland). The spectra were acquired and analyzed by WinQ and Easy View software (version 1.D, PerkinElmer, Turku, Finland).

The sodium salicylate was of 99% grade, purchased from HiMedia Laboratories Pvt. Ltd. (Mumbai, India).

### 2.2. Liquid Scintillation Counting Method and Equipment

^226^Ra content in water samples was tested via the commonly used LSC technique for ^222^Rn measurement. All LSC samples were prepared in High-Performance Glass vials from PerkinElmer (Turku, Finland), with a total volume of 20 mL, and the prepared samples were counted on an LS counter Quantulus 1220^TM^ (PerkinElmer, Turku, Finland).

The Quantulus spectrometer (PerkinElmer, Turku, Finland) has its own background reduction system around the vial chamber, which consists of a passive shield (lead, copper and cadmium) and an active shield as well. The mineral oil scintillator that surrounds the counting chamber presents the active shield of the instrument, since it has an additional pair of photomultiplier tubes that work in anticoincidence with the pair of photomultiplier tubes set around the counting chamber [19]. Low-activity materials were used in the construction of the Quantulus, which is an advantage when measuring low-level radiation activity. The Quantulus 1220^TM^ has two MultiChannel Analyzers (MCA), each divided into two halves. One MCA is used for the active shield, and the second is used for the spectra record. The system is provided with two pulse analysis circuits accessible tousers: a Pulse Shape Analysis (PSA) and Pulse Amplitude Comparator (PAC) circuit. PSA discriminates alpha- from beta-radiations and directs them separately into two MCA halves, alpha-MCA or beta-MCA [19].

#### Determination of ^226^Ra in Water by EPA Method 913.0

The physical basis for this method is the fact that, when mixed with a scintillation cocktail, radon from the water sample always diffuses into the organic phase for which it has a much greater affinity than for water [20]. For the calibration and standardization, the Radium Solution Method was applied, where a standard of 100 mL of ^226^Ra solution was prepared such that the final activity was ~1.3 kBq/L. According to EPA Method 913.0 [16], 10 mL of the diluted standard was transferred into a 20 mL scintillation vial, to which 10 mL of the scintillation cocktail had been added. The background samples were prepared using 10 mL of distilled water mixed with 10 mL of the same scintillation cocktail. The standards and the background samples were set aside for 30 days to allow radon to attain secular equilibrium. The samples were then counted for 50 min in an LS counter using an energy discrimination circuit for alpha/beta particles (PSA circuit). All LSC samples (10 mL of sample + 10 mL of scintillation cocktail) were prepared in 20 mL High-Performance Glass vials (Perkin Elmer) [20]. A selection of scintillation cocktails suitable for organic samples was tested for ^226^Ra determination: OptiPhase HiSafe 3 and Ultima Gold AB (water-miscible cocktails); Ultima Gold F, Mineral Oil Scintillator and OptiFluor O (water-immiscible cocktails).

For the optimal window selection, a radium standard was counted for 5 min. The region of the greatest alpha activity defined by two or three large peaks (generated by ^222^Rn, ^218^Po, ^214^Po) in the energy spectrum, varied in shape for different scintillation cocktails. The optimal window was formed according to the highest *FOM* value and was fixed for each of the cocktails used [21]. Verification of standard EPA Method 913.0 was done with a set of standard referent ^226^Ra sources.

The PSA parameter influence was investigated with the set of ^226^Ra standards for each of the cocktails used, and left for 1 month after preparation to reach radioactive equilibrium. PSA variation impacted the *CF* (Calibration Factor) value which then impacted the calculated ^226^Ra activity of the sample. It was determined that it was not necessary to set the PSA parameter at the crossover point (e.g., the least alpha/beta misclassification) before the sample measurement. The most important factor was to keep the PSA parameter fixed during *CF* determination and sample counting, in which case the PSA value itself did not influence ^226^Ra determination significantly [22].

Calibration factor (*CF*) [cpm/Bq] was calculated as follows [16]:(4)$$CF=\frac{S-B}{C_{0}\;V}$$
where *S* [cpm] and *B* [cpm] are the standard and the background count-rates, respectively, *C*_0_ [Bq/L] represents the concentration of ^226^Ra standard and *V* [L] is its total volume. The activity of ^226^Ra in the unknown water sample *C* [Bq/L] and its 2*σ* uncertainty (95% CI) were obtained using the following equations [16]:(5)CR226a=G−BCF V
(6)$$2\sigma =\frac{2\sqrt{\frac{G}{T_{G}}+\frac{B}{T_{B}}}}{CF\;V},$$
where *G* [cpm] is the gross count-rate of the sample, *V* [L] is the volume of sample, *T_B_* [min] and *T_G_* [min] represents the duration of the background and sample counting, respectively.

Evaluation of Minimum Detectable Activity (*MDA*) [Bq/L] for ^226^Ra in water samples could be carried out either via Currie relation, Equation (3) or based on the measurement uncertainty of the background, *u*(*B*) [cpm]:(7)$$MDA=\frac{4.65\sqrt{u\left(B\right)}}{CF\;V}$$

Equation (7) follows directly from Currie relation, Equation (3) when long and equal background and sample counting times are assumed.

### 2.3. Gamma Spectrometry Method and Equipment

All water samples were measured in a Marinelli beaker (0.5 L) without any chemical pretreatment by Canberra HPGe spectrometer, nominal efficiency of 35%, FWHM of 1.77 keV. A passive detector shield was made from 12 cm thick lead and an inner layer of 3 cm thick copper. The typical measurement time was 50,000 s. All measurement uncertainties are presented at a confidence level of 95% [23,24]. The most intensive post-radon lines of ^214^Pb and ^214^Bi (295.2 keV, 351.9 keV, 609.3 keV, 1120.3 keV) were used to calculate the ^226^Ra content of the samples. True coincidence corrections were applied to determine the activity from the ^214^Bi line. The gamma spectra were acquired and analyzed using the Canberra Genie 2000 software (Mirion Technologies Inc, Atlanta, GA, USA). The program calculates the activity concentration of an isotope from all prominent gamma lines after peaked background subtraction. The detector was calibrated using a standard reference radioactive material in a Marinelli beaker (multigamma standard resin matrix ^152^Eu, produced by FRAMATOM, France). Self-absorption effects due to different matrices/densities were taken into account using the efficiency transfer software ANGLE based on the concept of the effective solid angle [25,26]. During measurements, beakers were hermetically secured with tape.

### 2.4. Intercomparison Samples

A standard radioactive source, ^226^Ra, produced by Czech Metrology Institute, Inspectorate for Ionizing Radiation (Brno, Czech Republic), activity concentration of *C*_0_ = 39.67 Bq/mL on the reference date 1/10/2013, was used for the sample preparation. Samples were prepared with distilled water in 2 L plastic bottles, acidified with HNO_3_ and left in the laboratory for the time necessary radioactive equilibrium to be reached between ^222^Rn and ^226^Ra (~30 days). All measurements presented in the paper were carried out throughout the year 2017.

## 3. Results

### 3.1. Optimization of Cherenkov Counting Method

Cherenkov spectra are generated on Quantulus 1220^TM^ when the counting protocol is set up manually. The system configuration is explicitly displayed in the previous publication [18]. The counting was carried out on a high coincidence bias, since Cherenkov pulses are high amplitude pulses. The obtained spectral shapes of the ^226^Ra calibration sample (with and without sodium salicylate addition) are presented in Figure 1.

A significant number of Cherenkov photons cannot be detected by the photomultipliers placed in the LS counter since they are generated in the ultraviolet region. However, sodium salicylate added in a counting vial acts as awavelength shifter—it absorbs ultraviolet photons and re-emits them at longer wavelengths. This shift in wavelengths consequently leads to the detection of more photons and an increment in counting efficiency. In the case of ^210^Pb/^210^Bi detection, it was demonstrated that the addition of sodium salicylate > 1 mg/g increased detection efficiency due to the combination of wavelength shifting and the production of more scintillation light [14]. It is clear that the presence of sodium salicylate in ^226^Ra solution similarly generates a more intensive Cherenkov spectrum as seen in Figure 1.

The first step in method optimization was the selection of the optimal spectral window (ROI) which was carried out considering the count-rates of one vial with ^226^Ra solution and one background sample, requiring the maximal *FOM* [s] = *ε*^2^/*r*_0_, Figure Of Merit value. From the results in Table 1, one can observe that optimal ROI was established between 130 and 400 channels, since it provided maximal *FOM*, while *MDA* was as minimal as possible.

Calibration of the Quantulus detector facilitated determination of detection efficiency, which was experimentally assessed with an asset of five calibration samples, ^226^Ra solutions with increasing ^226^Ra activity. These samples were prepared in three probes containing distilled water and increasing amounts of the certified ^226^Ra activity, with a total volume of 20 mL in polyethylene vials. Each sample was counted in 6 cycles for 100 min. The obtained results are displayed in Figure 2a, where detection efficiency, 15.87 (24)%, was determined from the slope of the linear fit (with the intercept set to zero) of the data.

According to Equation (3), detection limits were evaluated while measurement duration was varied. *MDA* behavior is displayed in Figure 2b, and it can be observed that *MDA*~0.5 Bq/L (the limit set by the legislation in Serbia [6]) can be reached for 1000 min of counting.

Long counting times may be one of the drawbacks of the Cherenkov counting method, but there are ways to achieve lower *MDA*s with reduced measurement duration.

Small amounts of sodium salicylate added to the counting vial can significantly intensify the Cherenkov spectrum, as seen in Figure 1, meaning that the detection efficiency increases and *MDA* proportionally decreases. As shown in Figure 3, the effect of sodium salicylate addition to ^226^Ra calibration samples was tested. The graph from Figure 3a shows that an increasing mass of sodium salicylate added to the vials did not influence the count-rates of the background samples, therefore, the value *r*_0_ = 0.0045(10) cps can be regarded as constant. On the graph in Figure 3b, the consistency of the sodium salicylate addition was examined; a few samples were prepared with a different ^226^Ra activity into which the same mass of sodium salicylate, 0.2 g, was added. In all samples, the detection efficiency increased from 15.87 (24)% to the mean value 25.3 (20)%. These measurements indicate that sodium salicylate addition gives consistent results and provides stable samples, since the experimental results showed satisfactory repeatability. The addition of sodium salicylate to the counting vials during ^226^Ra measurement produced a similar effect on detection efficiency, as reported in the case of ^210^Pb/^210^Bi detection [14] and ^228^Ra/^228^Ac detection [13] via Cherenkov counting.

One other alternative for *MDA* decrement is to proportionally increase the sample volume that is being analyzed, which is evident in Equation (3). The radioactive content of the sample would be preconcentrated if the sample were evaporated before counting. In previous research, where ^210^Pb/^210^Bi detection via Cherenkov counting was explored, it was shown that a 10-fold reduction in *MDA* was accomplished when distilled water sample was evaporated from 200 mL to 20 mL, while the count-rate of a background sample was not altered [27].

The summation of the improvement of the method’s most important parameters is given in Table 2. One can conclude that the addition of 0.2 g of sodium salicylate to the counting vials ensures that *MDA* below the permitted levels in our country could be achieved during 500 min of counting. Moreover, in the case of the addition of 1 g of sodium salicylate to a ^226^Ra solution vial, the detection efficiency was enhanced to 38.3 (5)%, high enough that satisfactory *MDA* levels could be achieved for the purpose of routine radiological screening of samples. If lower *MDA* is required, the Cherenkov counting technique may have better sensitivity if the samples are evaporated before the counting. The combination of the sample’s evaporation from 200 mL to 20 mL, with the addition of 1 g of sodium salicylate, provided *MDA* = 0.025 Bq/L, which implies that the Cherenkov counting method has the potential for more sensitive radiological analysis of samples.

### 3.2. A Survey on Other Methods for ^226^Ra Determination

Before the evaluation of the EPA 913.0 method (“LSC-radon” technique) for the determination of ^226^Ra content in water samples, some experiments were conducted to test the performance of several scintillation cocktails for radium measurements. Ultima Gold AB and OptiPhase HiSafe 3 are water-miscible cocktails and form homogeneous samples, while Ultima Gold F, Mineral Oil and OptiFluor O are water-immiscible; from these, the two-phase samples were obtained. It was previously mentioned that the EPA 913.0 method is generally recommended for ^222^Rn activity concentration measurements, but it can be applied for ^226^Ra determination in samples that are known to have achieved secular equilibrium. For all of the used cocktails, calibration of the system was carried out according to the previously described procedure, which included *CF* determination, *CF* vs. PSA investigation, *MDA* calculation, etc. The majority of these measurements are reported in the previous research [21]. Results of measurements on the spiked ^226^Ra samples prepared with five different cocktails are displayed in Table 3. Samples mixed with Ultima Gold F and High-Performance Mineral Oil Scintillator cocktail (PerkinElmer) gave the best match concerning all referential ^226^Ra activity concentrations. High-Performance Mineral Oil Scintillator is often recommended for radon analysis. Optifluor O had the poorest performance in these experiments. The rest of the experiments concerning the comparison of the Cherenkov counting technique with the LSC-radon method were carried out only with the Ultima Gold F cocktail.

Table 4 shows the activity concentrations of the spiked samples obtained by three measurement techniques evaluated in this research.

First of all, the precision of the Cherenkov counting technique was investigated based on the measurement of the spiked samples, where the obtained activity concentrations had a relative deviation between 5and 20%. This indicates that Cherenkov radiation detection can provide a reliable estimation of ^226^Ra content in water samples. The results obtained via the other two commonly used methods offered the opportunity to compare those methods’ performances, advantages and drawbacks with those of the Cherenkov detection technique.

Results from gamma-spectrometric measurements have shown relative deviation <20%, except for the last sample with the greatest activity, for which the deviation was 69%. This discrepancy could be explained by the potential loss of ^222^Rn from the Marinelli beaker if the hermetical sealing was not perfect.

The obtained experimental results for ^226^Ra determination in the spiked water samples by LSC-radon method with Ultima Gold F scintillation cocktail had a relative deviation of 15% or less, which makes this method the most reliable for ^226^Ra measurements among the investigated methods.

With regard to detection limits, it can be observed in Figure 2b and Table 2 that the Cherenkov counting technique produced *MDA* = 0.415 Bq/L (slightly below the limit permitted by Serbian legislation [6]) over the total counting time of 1000 min, or for less than 500 min in the case of sodium salicylate addition (more precisely, it can be achieved in 300 min of counting when 1 g of sodium salicylate is added). For the purpose of comparison with other investigated methods, minimal detectable activity (*MDA*) vs. measurement time for ^226^Ra determination in water by: (a) gamma spectrometry, and (b) LSC-radon method when Ultima Gold F scintillator is used, is presented in Figure 4.

The presented graphs also contain equations the experimental data are fitted to. Distilled water was used as a blank sample for all investigated techniques. For the gamma-spectroscopic measurements, it was observed that *MDA* = 0.45 Bq/L when the measurement time was 20 h—therefore, gamma spectrometry offers the same *MDA* but for a longer measurement time in comparison with Cherenkov counting (without sodium salicylate). For the LSC-radon method. when samples were prepared with Ultima Gold F scintillator, *MDA* was calculated to be ~0.1 Bq/L for the total counting time of 300 min, which indicates that Cherenkov counting (without sodium salicylate) provides detection limits 10 times higher (or ~5 times higher in the case of sodium salicylate addition) than the LSC-radon method for the same measurement times.

The overall comparison of the advantages and disadvantages of all presented methods can be summarized in the conclusions presented in Table 5.

## 4. Discussion

Cherenkov’s counting technique offers satisfactory precision with deviations <20%. In its original form, it is a simple, non-expensive and non-destructive technique that offers reliable ^226^Ra screening in small-volumes water samples via LS counter. The purpose of this paper was to explore the Cherenkov counting technique on the Quantulus detector and the main parameters (efficiency, detection limit, reliability and precision) of the method. However, the most important drawback occurs if the sample contains other radionuclides besides ^226^Ra that can produce the Cherenkov spectrum, in which case, the interferences appear in the generated spectra, and chemical pretreatment is required before the counting. Further research should deal with the possible presence of ^40^K in groundwater (and, consequently, in drinking water) that can generate Cherenkov radiation and thus interfere with the ^226^Ra spectrum. A preconcentration step coupled with chemical separation should be carried out prior to counting to eliminate naturally occurring radionuclides, ^40^K or progenies from U and Th series [10]. Another drawback is the long counting time of 1000 min if endeavoring to reach the acceptable *MDA* parameter. However, the addition of 1 g of sodium salicylate ensures that detection limits below those legally permittedcan be reached within 300 min of counting. The investigation presented in this paper demonstrated that repeatable results are obtainable in the case of sodium salicylate addition.

Gamma spectrometry is a non-destructive and direct method that provided measurements of ^226^Ra content that were not reliable in higher activity concentrations (for *C*(^226^Ra) > 10 Bq/L). Its *MDA* is lower than the limit permitted by the legislation in Serbia [6], but it requires 20 h of measurement to achieve it. On the other hand, the LSC-EPA 913.0 method gave acceptable results on all ^226^Ra activity concentrations examined. This method is very simple, but it is destructive, since the sample is mixed with a scintillation cocktail. Very low detection limits can be achieved during measurements, which correspond to critical levels determined in studies [2,3,4] that impact human health. Precise and reliable results, the small sample volume required for the analysis and low *MDA* values make the LSC-radon method the most adequate among all those presented for medical research, epidemiologic studies and the dose assessment of drinking water.

## 5. Conclusions

Cherenkov counting technique for ^226^Ra determination in water was optimized on a Quantulus LS counter. The method was simple, rapid, and inexpensive. To date, the literature has not provided data on the Cherenkov counting technique for ^226^Ra determination on Quantulus. Therefore, the presented research enabled the assessment of the method’s potential in the field of routine radiological analysis of samples. The obtained detection efficiency was 15.87 (24)%, with a detection limit of *MDA* = 0.415 Bq/L for 1000 min of counting. The analyzed sample volume was 20 mL, and water samples were not pretreated. Long measurement time was shortened in the case of sodium salicylate addition, in which case 1 g of sodium salicylate increased efficiency to 38.1 (5)% with a reduction in the detection limit to *MDA* = 0.248 Bq/L for 500 min of counting. Sodium salicylate addition provided repeatable results. Additionally, if sample evaporation was performed before the counting, the detection limit was reduced proportionally to volume reduction during evaporation. Cherenkov counting offered reliable ^226^Ra measurement with satisfactory precision, since relative deviation was less than 20%. Although Cherenkov counting is a very efficient screening tool for ^226^Ra determination, the problem of interferences with other radionuclides capable of generating Cherenkov photons, such as ^40^K, should be addressed in future work.

A brief comparison of the Cherenkov counting method with two other common measurement methods for the determination of ^226^Ra activity concentrations in water was also conducted. *MDA* determined via gamma spectrometry was 0.45 Bq/L for 20 h of measurement, and the activity concentrations of spiked samples had larger deviations than the ones obtained by the Cherenkov counting technique. The LSC-radon measurement technique resulted in *MDA* = 0.1 Bq/L for 300 min of measurement, with relative deviations less than 15%. It can be concluded that the most precise results for ^226^Ra measurements were yielded by the LSC-radon method, which is relatively fast, does not require any chemical pretreatment of samples, and requires a sample volume of only 10 mL. The advantage of the Cherenkov counting technique in comparison with the LSC-radon method lies in the fact that no scintillation cocktail is needed for the sample analysis, which makes this technique, besides being less expensive and non-destructive, also environmentally friendly and safe from the ecological perspective, with regard to the storage of samples after their counting.

## Figures and Tables

**Figure 1 materials-14-06719-f001:**
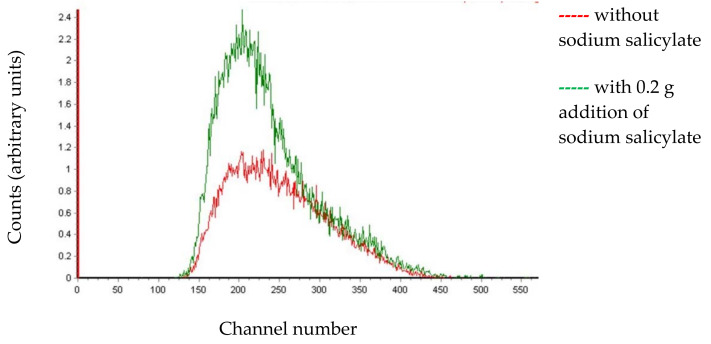
Generated Cherenkov spectra of ^226^Ra solution (*C*_0_ = 15.87 Bq).

**Figure 2 materials-14-06719-f002:**
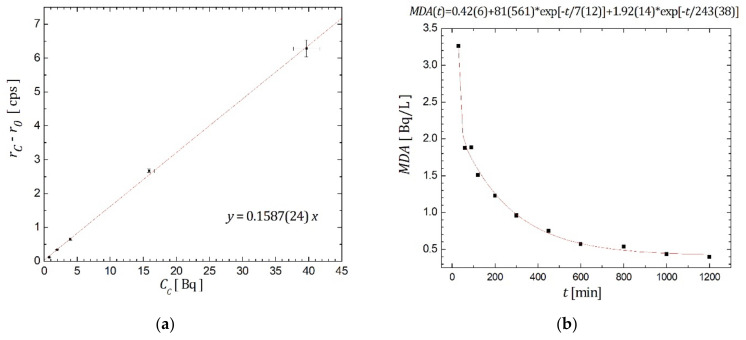
Cherenkov counting technique optimization for ^226^Ra determination in water: (**a**) Results of the calibration procedure; (**b**) *MDA* achieved.

**Figure 3 materials-14-06719-f003:**
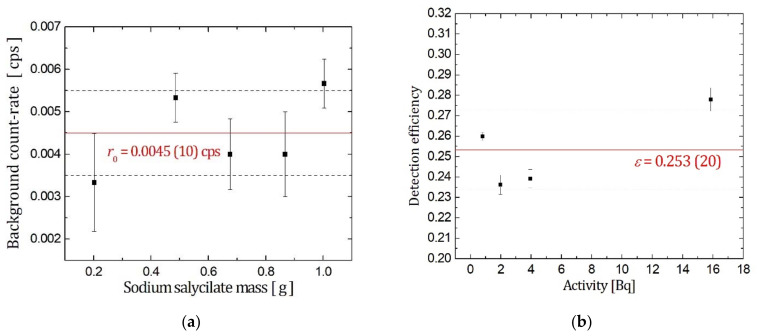
The addition of sodium salicylate: (**a**) increasing amounts added to the background samples; (**b**) 0.2 g added to ^226^Ra solution samples with different activities.

**Figure 4 materials-14-06719-f004:**
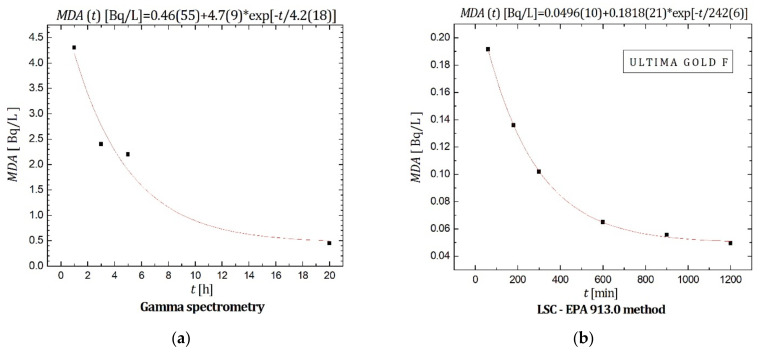
*MDA* for ^226^Ra determination in water by: (**a**) gamma-spectrometry; (**b**) LSC-radon method using Ultima Gold F scintillator.

**Table 1 materials-14-06719-t001:** Optimal spectral window selection (fine-tuning of ROI).

Spectral Window [Channels]	*r*_0_ [cps]	*ε*	*MDA* [Bq/L]	*FOM* [s]
130–430	0.0070 (4)	0.171 (6)	0.894	4.2 (4)
130–410	0.0070 (4)	0.170 (6)	0.898	4.1 (4)
130–400	0.0060 (4)	0.169 (6)	0.838	4.8 (5)
140–410	0.0070 (4)	0.170 (6)	0.899	4.1 (4)
150–410	0.0070 (4)	0.168 (6)	0.906	4.1 (4)
150–420	0.0070 (4)	0.169 (6)	0.903	4.1 (4)
130–390	0.0060 (4)	0.168 (6)	0.843	4.7 (5)

**Table 2 materials-14-06719-t002:** Characteristic parameters in the case of the method’s improvement.

	Without Sodium Salicylate	0.2 g of Sodium Salicylate Addition	1 g of Sodium Salicylate Addition	1 g of Sodium Salicylate Addition + Evaporation 200 mL to 20 mL
***ε* [%]**	0.1587 (24)	0.253 (20)	0.381 (5)	0.381 (5)
***MDA* [Bq/L]** ***t*_0_ = 500 min**	0.596	0.373	0.248	0.025
***MDA* [Bq/L]** ***t*_0_ = 1000 min**	0.415	0.260	0.173	0.017

**Table 3 materials-14-06719-t003:** Results were obtained on the intercomparison samples prepared with different scintillation cocktails used with the LSC-radon method.

Reference *C*_0_ [Bq/L]	*C*_exp_ [Bq/L] Ultima Gold AB	*C*_exp_ [Bq/L] Ultima Gold F	*C*_exp_ [Bq/L] OptiPhase HiSafe 3	*C*_exp_ [Bq/L] OptiFluor O	*C*_exp_ [Bq/L] Mineral Oil
0.3970 (20)	0.35 (7)	0.35 (7)	0.340 (25)	<*MDA*	0.44 (6)
1.587 (8)	1.53 (21)	1.57 (22)	1.30 (10)	0.77 (8) *	1.7 (4)
3.966 (20)	3.03 (19)	3.39 (27)	4.4 (3)	3.08 (25)	3.7 (4)
7.93 (3)	7.56 (24)	7.4 (3)	6.2 (3)	5.1 (3)	7.8 (5)
9.92 (5)	7.8 (4)	8.6 (5)	8.0 (7)	7.3 (8)	8.7 (5)

* This sample showed an increased level of quench, which caused a spectral shift so that part of the spectrum was placed outside the optimal counting window.

**Table 4 materials-14-06719-t004:** Results of intercomparison.

Reference *C*_0_ [Bq/L]	Cherenkov Counting *C*_exp_ [Bq/L]	LSC-Radon *C*_exp_ [Bq/L]	Gamma-HPGe *C*_exp_ [Bq/L]
0.3970 (20)	0.35 (14)	0.35 (7)	0.40 (24)
1.587 (8)	1.26 (24)	1.57 (22)	1.3 (7)
3.966 (20)	4.2 (4)	3.39 (27)	3.6 (16)
5.95 (3)	5.6 (4)	6.6 (10)	5.9 (16)
7.93 (3)	7.2 (5)	7.4 (3)	8.1 (14)
9.92 (5)	9.4 (5)	8.6 (5)	16.8 (9)

**Table 5 materials-14-06719-t005:** Characteristic parameters for the presented methods (the untreated water samples).

	Cherenkov Counting	Gamma Spectrometry	LSC-Radon
**Sample volume [mL]**	3 × 20	450	3 × 10
**Counting time [min**]	300	1200	300
***MDA* [Bq/L]**	~1 (original method) ~0.5 (with 1 g sodium salicylate)	0.45	~0.1

## Data Availability

The data presented in this study are available on request from the corresponding author.

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
