# Peer review of "Cherenkov Radiation Detection on a LS Counter for 226Ra Determination in Water and Its Comparison with Other Common Methods"

_materials, 2021, doi:10.3390/ma14216719_

Round 1

Reviewer 1 Report

The research presented in this work provides a very nice overview of the different methods used to determine the radioactivity in drinking water. The authors present a rigorous methodology for their experiments, and provide useful results in the form on comparative methods of different techniques that can be used to determine the radioactivity. A few minor edits should be made prior to publication:

There are instances in the manuscript where there should be more care applying spacing between words, e.g., '40Kin' on Line 373. 

Also, if possible, high-resolution figures would be better suited for journal publication. 

Author Response

Comments and Suggestions for Authors

The research presented in this work provides a very nice overview of the different methods used to determine the radioactivity in drinking water. The authors present a rigorous methodology for their experiments, and provide useful results in the form on comparative methods of different techniques that can be used to determine the radioactivity. A few minor edits should be made prior to publication:

There are instances in the manuscript where there should be more care applying spacing between words, e.g., '40Kin' on Line 373. 

Also, if possible, high-resolution figures would be better suited for journal publication. 

Response. We thank the reviewer for his/her evaluation of our research. We also thank for the comment about the spacing between words, we found indeed few more spacing-missing points in the text and we corrected all those. All changes can be seen via “Track Changes” in the revised version of the manuscript. As for the resolution of figures, we have changed it to higher resolution as suggested, from 300 dpi to 500 dpi.

Reviewer 2 Report

The author presented the application of Cerenkov counting method to detect 226Ra in the water samples and the use of the chemical, sodium salicylate to increase the counting efficiency, resulting in the decrease of the MDA. And, the author carried out the experiments to discuss the comparison between various methods including the LSC counting method using the scintillation cocktail, Gamma spectrometry method in terms of the sample volume, counting time, and MDA.

However, the reviewer recognized that the author's way is not the effective way for lowering MDA. As the author presented in this article, the most effective way to decrease MDA for the detection of 226Ra in the water sample is the decrease of the sample volume through evaporation. If the volume of the water samples is concentrated from 200 mL to 20 mL by evaporation, the MDA value is decreased by 10 times. In addition, it does not take a long time to evaporate the water of 200 mL.

So, it is difficult to find the scientific value of this article in the field of the radioactive measurement field.

And, the reviewer gives some comments.

1) In table 1.

 The reviewer does not agree that the results are really meaningful in this study. The author showed various ranges of spectral window, but the most difference of MDA in the conditions is less than just 10 %.  It is more effective to compare it in 100-450 rather than 130-410 to highlight 130-400 for the optimal spectral channel.

2) In figure 3.

 The author tried to discuss the addition of sodium salicylate contributes to the increase of the counting efficiency. It seems to be clear that the counting efficiency is increased from 15.8% to 25.3% due to sodium salicylate. However, figure 3 does not effectively show the increase in counting efficiency with the use of the reagent because the range of the y-axis is too small.

3) Table 2

Table 2 explicitly shows that the evaporation of the water sample is the most effective way to lower the MDA level.

4) the presence of the interferences

It is recommended that the author carry out some experiments to estimate the potential interferences, for example, 40K in the water.

The author presented the application of Cerenkov counting method to detect 226Ra in the water samples and the use of the chemical, sodium salicylate to increase the counting efficiency, resulting in the decrease of the MDA. And, the author carried out the experiments to discuss the comparison between various methods including the LSC counting method using the scintillation cocktail, Gamma spectrometry method in terms of the sample volume, counting time, and MDA.

However, the reviewer recognized that the author's way is not the effective way for lowering MDA. As the author presented in this article, the most effective way to decrease MDA for the detection of 226Ra in the water sample is the decrease of the sample volume through evaporation. If the volume of the water samples is concentrated from 200 mL to 20 mL by evaporation, the MDA value is decreased by 10 times. In addition, it does not take a long time to evaporate the water of 200 mL.

So, it is difficult to find the scientific value of this article in the field of the radioactive measurement field.

And, the reviewer gives some comments.

1) In table 1.

 The reviewer does not agree that the results are really meaningful in this study. The author showed various ranges of spectral window, but the most difference of MDA in the conditions is less than just 10 %.  It is more effective to compare it in 100-450 rather than 130-410 to highlight 130-400 for the optimal spectral channel.

2) In figure 3.

 The author tried to discuss the addition of sodium salicylate contributes to the increase of the counting efficiency. It seems to be clear that the counting efficiency is increased from 15.8% to 25.3% due to sodium salicylate. However, figure 3 does not effectively show the increase in counting efficiency with the use of the reagent because the range of the y-axis is too small.

3) Table 2

Table 2 explicitly shows that the evaporation of the water sample is the most effective way to lower the MDA level.

4) the presence of the interferences

It is recommended that the author carry out some experiments to estimate the potential interferences, for example, 40K in the water.

Reviewer 3 Report

Please improve the state of the art analysis to clearly show the progress beyond the state of the art. The main novelty in this work must be clearly pointed out. The authors need to emphasize the research novelty, research significance, and contributions to academics and practices. The innovation and the importance of this work are not clearly highlighted in the abstract, introduction, and conclusions. Please work on this and prove to us why this work is valuable. The structure of the work needs to reorganize. Discussion of the results should provide useful insights. This manuscript lacks sufficient scientific novelty. So, the author(s) should be clearer about the uniqueness of the study. 

  1. Abstract need to rewrite. More detail should be included. The abstract should be rewritten by adding more informative data and results. Data should be incorporated into the abstract. The standard format of the Abstract: aim, background, motivation, hypothesis, methods, results and conclusions were specified more clearly. Please break some of the sentences. In the abstract, please add an indication of the achievements of your study that are relevant to the journal scope. Please be concise - maximum 1-2 lines.
  2. The introduction section should follow the state of the art of this field and review what has been done, for supporting the research gap and the significance of this study. Please improve the state of the art overview, to clearly show the progress beyond the state of the art. The lack of proper justification creates the wrong impression that the authors are unaware of the recent developments. Please also remove any lumped references. Please define each of them separately to avoid inappropriate citations. At the end of the introduction, the statement of the paper goal and the explanation of novelty has to be properly formulated.
  3. Results and Discussion: this section should be rewritten by adding more informative data and results. The authors should perform a comparison between the forecasting results with those of the literature. The explanation of the results should be elaborated. What can you conclude from the results? What does it mean? You should explain results in relation to your research questions, then show their implication to the research. All the obtained results need to discuss along with the findings of other researchers.

  4. Conclusions must go deeper. The conclusions can still be improved by providing an analysis of where the current work on adsorbents is focused, what are the remaining gaps in literature where more research should be conducted. In addition to summarizing the actions taken and results, please strengthen the explanation of their significance. Also, you need to explain the research novelty, limitation of the study, and contributions of the study for academics and practices, particularly for cleaner production. Quantitative results are also required in this section.

    Bibliography style is not always consistent, please double-check the reference section carefully and correct the inconsistency. They need to strictly follow the journal's guidelines.

Round 2

Reviewer 3 Report

Good paper.